# SPIN90 Deficiency Ameliorates Amyloid β Accumulation by Regulating APP Trafficking in AD Model Mice

**DOI:** 10.3390/ijms231810563

**Published:** 2022-09-12

**Authors:** Youngsoo Oh, Wongyoung Lee, So Hee Kim, Sooji Lee, Byeong C. Kim, Kun Ho Lee, Sung Hyun Kim, Woo Keun Song

**Affiliations:** 1Cell Logistics Research Center, School of Life Science, Gwangju Institute of Science and Technology, Gwangju 61005, Korea; 2Department of Neuroscience, Graduate School, Kyung Hee University, Seoul 02447, Korea; 3Department of Medicine, School of Medicine, Kyung Hee University, Seoul 02447, Korea; 4Department of Neurology, Chonnam National University Medical School, Gwangju 61469, Korea; 5Gwangju Alzheimer’s Disease and Related Dementia Cohort Research Center, Chosun University, Gwangju 61452, Korea; 6Department of Physiology, School of Medicine, Kyung Hee University, Seoul 02447, Korea

**Keywords:** SPIN90, Alzheimer’s disease, Amyloid β accumulation, APP trafficking, APP recycling, Rab11, synaptic transmission

## Abstract

Alzheimer’s disease (AD), a common form of dementia, is caused in part by the aggregation and accumulation in the brain of amyloid β (Aβ), a product of the proteolytic cleavage of amyloid precursor protein (APP) in endosomes. Trafficking of APP, such as surface-intracellular recycling, is an early critical step required for Aβ generation. Less is known, however, about the molecular mechanism regulating APP trafficking. This study investigated the mechanism by which SPIN90, along with Rab11, modulates APP trafficking, Aβ motility and accumulation, and synaptic functionality. Brain Aβ deposition was lower in the progeny of 5xFAD-SPIN90KO mice than in 5xFAD-SPIN90WT mice. Analysis of APP distribution and trafficking showed that the surface fraction of APP was locally distinct in axons and dendrites, with these distributions differing significantly in 5xFAD-SPIN90WT and 5xFAD-SPIN90KO mice, and that neural activity-driven APP trafficking to the surface and intracellular recycling were more actively mobilized in 5xFAD-SPIN90KO neurons. In addition, SPIN90 was found to be cotrafficked with APP via axons, with ablation of SPIN90 reducing the intracellular accumulation of APP in axons. Finally, synaptic transmission was restored over time in 5xFAD-SPIN90KO but not in 5xFAD-SPIN90WT neurons, suggesting SPIN90 is implicated in Aβ production through the regulation of APP trafficking.

## 1. Introduction

Alzheimer’s disease (AD), the most prevalent type of neurodegenerative disease, is caused by several factors, including amyloid β (Aβ) oligomer deposition and Tau based neurofibrillary tangles [1]. Aβ peptides accumulate in extracellular spaces and form Aβ aggregates, which are toxic to neurons and cause synaptic dysfunction, memory loss and cognitive impairments [2]. Aβ peptides are produced by the sequential cleavage of amyloid β precursor protein (APP) by the enzymes β-site APP-cleaving enzyme 1 (BACE1) and γ-secretase complex [3]. Both APP and BACE1 are transmembrane proteins internalized into early endosomes, with γ-secretase complex being later internalized into endosomes [4]. Endosomal APP is cleaved by BACE1 and γ-secretase, producing Aβ peptides [5,6].

APP is a transmembrane protein consisting of a long ectodomain, a glycosylation domain and a short cytoplasmic domain. Because Aβ peptide is generated from APP, studies have focused on APP processing [7,8]. However, proper APP trafficking is required for APP processing [6,9,10,11]. For example, cotrafficking of APP and BACE1 is essential for APP processing, which is induced by neural activities [12,13]. Rab35, a member of the Rab-GTPase protein family involved in membrane trafficking, was shown to negatively regulate Aβ production by coordinating the intracellular trafficking of APP and BACE1 [14]. RNAi screening of all Rab-GTPase proteins identified Rab3 and Rab11 as key molecules involved in the amyloidogenic process [15]. Furthermore, this trafficking was accelerated by mutant forms of APP [16,17]. Less is known, however, about the general distribution of APP on the surfaces and in the internal areas of neurons, such as dendrites and axons; about APP recycling trafficking; or about the molecular mechanisms and molecules involved in this process.

SH3 protein interacting with Nck, 90 kDa (SPIN90) is a protein interacting with the non-catalytic region of tyrosine kinase (Nck) [18] that participates in various membrane trafficking processes, including membrane ruffling, clathrin-mediated endocytosis, epidermal growth factor receptor (EGFR) endocytosis and endosomal trafficking [19,20,21,22,23]. By interacting with syndapin and dynamin1, SPIN90 is involved in clathrin-mediated endocytosis. The ablation of SPIN90 was found to attenuate the trafficking of early endosomes [23]. SPIN90 was also found to interact with Rab5 (early endosome regulator) and Gapex5 (Rab5 GEF) during epidermal growth factor (EGF) mediated endocytosis [24]. These findings suggested that SPIN90, along with Rab GTPase, is involved in Aβ production through its regulation of APP trafficking.

The present study describes the development of a mouse model in which 5xFAD mice were crossed with SPIN90 wild-type (WT) and SPIN90 knockout (KO) mice, with the SPIN90 KO mice showing improved Aβ deposition in mouse brains. SPIN90 was found to regulate the distribution of APP on the surfaces and internals of axons and dendrites, to modulate the activity-dependent recycling of APP between the surfaces and internals and to control axonal trafficking of APP. SPIN90 KO was found to reduce anterograde APP motility and its internal accumulation, and SPIN90 was found to interact with Rab11. Finally, SPIN90 ablation in 5xFAD neurons restored synaptic functionality. Collectively, these findings indicate that SPIN90 modulates APP trafficking along with Rab11, thereby controlling Aβ production.

## 2. Results

### 2.1. Deficiency of SPIN90 Reduces Aβ Deposition in the Brains of AD Model Mice

The involvement of SPIN90 in membrane trafficking [21,22,23,24] and the importance of APP trafficking in the early process of Aβ generation suggested that SPIN90 may be related to the pathogenesis of AD, particularly the production of Aβ. To determine whether SPIN90 interacts genetically with Aβ deposition, 5xFAD mice, a model of AD characterized by the rapid accumulation of Aβ plaques due to the expression of APP mutant transgenes [25,26], were mated with SPIN90 KO mice, which have been used in functional studies of membrane trafficking and neural function [24,27]. The time course of Aβ deposition in the hippocampus and subiculum was compared immunohistochemically in 5xFAD-SPIN90 WT (5xFAD/SPIN90+/+) and 5xFAD-SPIN90 KO (5xFAD/SPIN90−/−) mice. Anti-Aβ antibody (4G8) was applied to brain slices of mice aged 2, 3, 4, and 6 months, with binding visualized using the Dab staining system. As expected, the deposition of Aβ in the hippocampus and subiculum significantly increased over time in 5xFAD-SPIN90 WT mice, but was relatively lower in 5xFAD-SPIN90 KO mice (Figure 1A–D). Further evaluation of Aβ plaque accumulation in brain slices by thioflavin staining showed that the levels of Aβ plaque were lower in the hippocampus (~57%) and subiculum (~31%) of 5xFAD-SPIN90 KO than of 5xFAD-SPIN90 WT mice (Figure 1E,F) and that the mean sizes of Aβ plaques were slightly decreased in the hippocampus (~18.5%) and subiculum (~23%) of 5xFAD-SPIN90 KO (Appendix A). Assays to determine whether decreased Aβ deposition was caused by decreased APP expression in 5xFAD-SPIN90 KO mouse brains showed that the levels of expression of APP, as well as of related proteins such as BACE1 and nicastrin, did not differ in 5xFAD-SPIN90 KO and 5xFAD-SPIN90 WT mice (Appendix A), nor did the activity of BACE1 (Appendix A). Thus, these results suggest that SPIN90 deficiency down-regulates Aβ production and accumulation.

### 2.2. APP Distribution on the Surfaces of Axons and Dendrites Is Altered in 5xFAD-SPIN90 KO Neurons

Findings showing that the deletion of SPIN90 reduced Aβ production and that SPIN90 is involved in membrane trafficking [21,22,23,24] suggested that SPIN90 may be involved in APP trafficking. To test this hypothesis, APP distribution was evaluated on the membranes of various areas of neurons, including axons, dendrites, and soma. pHluorin has a pKa of 7.1, making it useful for evaluating surface and internal vesicle trafficking [28,29]. Surface and internal APP were therefore traced using a recombinant APP with pHluorin at its N-terminus (pH-APP) (Figure 2A). APP on membrane surfaces and internals could be quantified by sequentially applying acid quenching solution (AQ) (e.g., MES, pH5.5) and alkalizing solution (e.g., NH_4_Cl, pH7.4) [30]. Under resting conditions, the surface fraction of APP differed in various areas of 5xFAD-SPIN90 WT neurons, with higher accumulation in dendrites and soma (about 10–12%) than in axons (about 5.85%) (Figure 2B–H, Appendix A). APP distribution differed, however, in the 5xFAD-SPIN90 KO neurons, with higher APP surface fraction in dendrites (~23.3%) and lower accumulation in axons (~2.7%) (Figure 2B–H). These findings indicate that the surface distribution of APP is both locally distinct and regulated by SPIN90.

### 2.3. SPIN90 Modulates Activity-Dependent APP Trafficking between Surfaces and Internals of Axons and Dendrites

To determine whether SPIN90 modulates activity-dependent APP trafficking, 5xFAD-SPIN90 WT and 5xFAD-SPIN90 KO neurons transfected with pH-APP were electrically stimulated with 20 Hz for 30 s, and APP trafficking in each region (i.e., axons, dendrites, and soma) was monitored. Region-specific trafficking of APP was observed in 5xFAD-SPIN90 WT neurons. In axons, APP surface level was increased in response to electrical stimulation (Figure 3A–D), indicating that internal APP is exocytosed during stimulation, and endocytosed after the end of stimulation. In contrast, surface APP in dendrites was endocytosed during electrical stimulation, and recycled after stimulation, as previously reported [12] (Figure 3E–H), with surface APP in soma showing a similar response to electrical stimulation as in dendrites, being endocytosed during neuronal activity (Appendix A). In addition, ablation of SPIN90 neurons significantly enhanced activity-dependent APP trafficking. Activity-dependent surface accumulation of APP was ~2.5 fold higher on the surfaces of axons (Figure 3C,D and see also Appendix A) and ~2.8 fold higher on the surfaces of dendrites (Figure 3G,H and see also Appendix A) than on the corresponding surfaces of 5xFAD-SPIN90 WT neurons, with findings in soma similar to those in dendrites (Appendix A). The rates of exo- and endocytosis of APP were also determined by analyzing slopes of exo- and endocytosis. The slopes of exocytosis during neuronal activity and of endocytosis after stimulation were ~2.5-fold higher in axons of 5xFAD-SPIN90 KO than of 5xFAD-SPIN90 WT neurons (Figure 3I–L). Conversely, the slopes of endocytosis during stimulation and of exocytosis after stimulation were about 3-fold higher in dendrites of 5xFAD-SPIN90 WT than of 5xFAD-SPIN90 KO neurons (Figure 3M–P), with activity-dependent APP trafficking in soma similar to that in dendrites (Appendix A). To determine whether these phenotypes of APP trafficking are caused by artifacts that are dependent on the parallel accumulation of APP in axons or its diffusion in dendrites, changes in fluorescence in regions of interest (ROIs) and adjacent areas were carefully analyzed in axons and dendrites. During stimulation, the intensity of fluorescence was increased in ROIs of axons and was modestly decreased in areas adjacent to these ROIs, whereas the intensity of fluorescence was decreased in ROIs of dendrites and was modestly decreased in areas adjacent to these ROIs, indicating that APP is recycled (exo-endo) in an activity-dependent manner, not through accumulation or diffusion by parallel movements (Appendix A). Collectively, these findings indicate that, during neuronal activity, the pathways for APP trafficking between surfaces and internal areas are locally distinct, with APP exocytosis in axons and endocytosis in dendrites, and that SPIN90 markedly modulates activity-driven, locally distinctive APP trafficking in axons and dendrites (Figure 3Q,R).

### 2.4. SPIN90 Is Implicated in the Axonal Transport of APP

APP processing and trafficking can be further classified not only by local recycling between surfaces and internals but also by the motility of APP via axons and dendrites. To determine whether SPIN90 is involved in APP motility and trafficking via axons, neurons cotransfected with GFP-APP and RFP-SPIN90 were monitored by live-cell imaging. This monitoring showed that some GFP-APP and RFP-SPIN90 molecules colocalize and remain stationary whereas other GFP-APP and RFP-SPIN90 molecules move together through axons (Figure 4A). Analysis of APP motility in 5xFAD-SPIN90 WT and 5xFAD-SPIN90 KO neurons showed that APP motility through axons was lower in 5xFAD-SPIN90 KO than in 5xFAD-SPIN90 WT neurons, with anterograde movement also reduced in 5xFAD-SPIN90 KO neurons (Figure 4B–F), indicating that SPIN90 is involved in APP motility through axons.

### 2.5. SPIN90 Regulates Internal APP Accumulation in Axons

The involvement of SPIN90 in APP motility through axons, coupled with the motility associated aggregation of GFP-APP signals suggested that the latter may be associated with increases in stationary and retrograde motility of APP in axons and the accumulation of APP with presynaptic proteins in the brains of patients with AD [12]. This hypothesis was tested by monitoring APP aggregation and assessing whether this accumulation is internal or on the membrane surface. Neurons from both 5xFAD-SPIN90 WT and 5xFAD-SPIN90 KO mice expressing pH-APP were evaluated after the sequential application of AQ and NH_4_Cl solutions. APP was found to accumulate in internal areas, but not on membrane surfaces (Figure 5A,B). In addition, the percentage of neurons without internal APP accumulation was more than three-fold higher in 5xFAD-SPIN90 KO (~43%) than in 5xFAD-SPIN90 WT (~9%) mice (Figure 5C). Among neurons positive for APP, the amount and size of APP accumulation were significantly lower in 5xFAD-SPIN90 KO than in 5xFAD-SPIN90 WT neurons (Figure 5D–G). These findings indicate that, in the absence of SPIN90, APP axonal trafficking is decreased or becomes more the retrograde, reducing internal APP accumulation in axons and suggesting that SPIN90 is a regulator of APP trafficking and accumulation. Notably, we cannot find these accumulations of APP in the dendrite area (Appendix A) [31].

### 2.6. SPIN90 Preferentially Interacts with the Inactive form of Rab11 GTPase

SPIN90 has been shown to participate in Rab protein mediated membrane trafficking by enhancing Rab activity [23,24], and Rab11 was identified as a regulator of Aβ production [15]. These findings suggested that modulation of APP trafficking by SPIN90 may be associated with Rab11. The ability of SPIN90 to physically interact with Rab11 was tested by in situ and biochemical interaction assays. In the in-situ proximity ligation assays (PLA), control or SPIN90-specific shRNA was introduced into cells, followed by incubation with antibodies specific to SPIN90 and Rab11 and monitoring using the Duolink visualization system. Numerous interacting puncta were observed in control cells, with only about one third the number of puncta observed in SPIN90 knock-down cells (Figure 6A,B). An in vitro binding assay showed that the C-terminus of SPIN90 interacts directly with and binds to Rab11 (Figure 6C). To determine whether its interaction with SPIN90 is related to the status of Rab11 activity, the ability of SPIN90 to interact with active and mutated, inactive forms of Rab11 was evaluated by in vitro binding assays. SPIN90 was found to preferentially interact with an inactive form of Rab11, Rab11 S25N (Figure 6D–G), increasing stationary APP in axons (Appendix A), suggesting that the interaction of SPIN90 with Rab11 may contribute to APP trafficking. 

Furthermore, the distribution of SPIN90, APP and Rab11 in the synapses of neurons was also evaluated. The three molecules were coexpressed in various combinations along with VAMP2, a presynaptic marker, and their colocalization in nerve terminals was analyzed. SPIN90 and Rab11 were highly colocalized at synapses (more than 80%), as were SPIN90 and APP (about 50%) (Figure 6H–W), suggesting that SPIN90-Rab11 may cooperate in regulating APP trafficking in neurons.

### 2.7. SPIN90 Deficiency Restores Activity-Driven Synaptic Function in AD Model Neurons

Although AD usually results in neurodegeneration or neuronal cell death, Aβ has been shown to cause synaptic depression and synaptopathy [32,33,34,35,36] prior to neuronal cell death. Because SPIN90 ablation reduced Aβ accumulation, the ability of SPIN90 deficiency to restore activity-driven synaptic function, such as synaptic transmission, was investigated. The combination of a primary hippocampal culture system and a pHluorin-based assay has been utilized to monitor activity-driven synaptic transmission [37,38]. The fusion of the pH-sensitive GFP protein pHluorin to the luminal region of the synaptic vesicle protein, vesicular glutamate transporter1 (vGlut1), yields the protein vGlut-pHluorin (vG-pH), a very efficient sensor of synaptic function (e.g., synaptic transmission) [28,30]. The protein vG-pH was introduced into primary hippocampal neurons from WT-SPIN90 WT, WT-SPIN90 KO, 5xFAD-SPIN90 WT, and 5xFAD-SPIN90 KO mice, and synaptic transmission in response to stimulation with 10 Hz for 10 s (100 APs) was monitored. Synaptic transmission was ~50% lower in 5xFAD-SPIN90 WT than in WT-SPIN90 WT neurons (Figure 7A–C), in good agreement with previous results [34]. However, synaptic transmission in 5xFAD-SPIN90 KO neurons recovered almost completely, similar to that in controls (Figure 7A–C). These findings suggested that the reduced level of Aβ plaque accumulation in 5xFAD-SPIN90 KO neurons has an effect on synaptic functionality.

To closely mimic neurodegeneration in humans, synaptic functionality over time was evaluated in the four types of hippocampal neurons. As expected, synaptic transmission in 5xFAD-SPIN90 WT neurons was gradually reduced over time, whereas synaptic transmission in 5xFAD-SPIN90 KO neurons remained unchanged, with a functionality of up to ~20% (Figure 7D–J), suggesting that APP trafficking and processing via the SPIN90 pathway is related to synaptic functionality.

## 3. Discussion

The involvement of Aβ in the pathology of AD has suggested that APP processing is crucial for the development of AD. During the amyloidogenic process, APP is cleaved by BACE1 and γ-secretase to yield Aβ which accumulates in brain tissue. Thus, the spatiotemporal regulation of APP and these enzymes is involved in the development of AD. In addition to controlling the activity of these enzymes, the regulation of APP trafficking is a critical process in the amyloidogenesis pathway because the availability of APP controls both enzyme activity and Aβ production. 

The present study found that the depletion of SPIN90 from 5xFAD AD model brains reduced Aβ deposition and restored synaptic functionality, suggesting that SPIN90 may be involved in the production of Aβ from APP. Moreover, the surface and internal distribution of APP differed in axons and dendrites, with SPIN90 deficiency significantly altering these distributions. For example, the amount of surface APP was lower in axons but higher in dendrites of resting 5xFAD-SPIN90 KO than resting 5xFAD-SPIN90 WT neurons. The activity-driven recycling of APP on the surfaces and internal areas of both axons and dendrites was significantly higher in 5xFAD-SPIN90 KO than in 5xFAD-SPIN90 WT neurons. The finding, that SPIN90 interacts with inactive Rab11 protein, suggests that the regulation of APP trafficking might be coordinated by complexes of SPIN90 and Rab11. SPIN90-Rab11 and SPIN90-APP were found to colocalize at nerve terminals. This study also found that SPIN90 is cotrafficked with APP and that, in the absence of SPIN90, the transport of APP and its internal accumulation in axons were reduced. Collectively, these results indicate that SPIN90, together with Rab11, regulates APP trafficking and may affect the efficiency of Aβ production. 

These findings raise several intriguing points regarding APP trafficking, including whether the localized distributions of APP in axons and dendrites play different roles in APP processing. The distribution of APP in axons and dendrites and its local recycling between cell surfaces and internal areas are distinct. The surface distribution of APP is not the same on axons and dendrites of 5xFAD mouse neurons, with axons having about 5% and dendrites having about 10% surface APP. This difference was increased in the absence of SPIN90, with axons having about 2.5% and dendrites having about 25% surface APP, suggesting that SPIN90 controls the specific subcellular distribution of APP in neurons. In addition, activity-driven APP recycling between surfaces and internal areas differed significantly in axons and dendrites. During activation, intracellular APP in axons moved to the surface, but was endocytosed after stimulation. In dendrites, surface APP was endocytosed during neuronal stimulation, but was subsequently recycled to the surface after stimulation. These phenotypes were strongly enhanced in the absence of SPIN90, suggesting that SPIN90 modulates activity-dependent local APP recycling. Moreover, the involvement of Rab11 suggests that the interaction between SPIN90 and the inactive form of Rab11 may be involved in the distribution and recycling of APP. Because a negative form of Rab11 downregulates Aβ production [15], SPIN90 may regulate Rab11 guanine nucleotide exchange factor (GEF) or act as a cofactor of GEF during Rab11-mediated amyloidogenesis.

Another question is related to internal APP accumulation in axons. APP was shown to accumulate in the brains of patients with AD [31]. The present study found that SPIN90 was partially cotrafficked with APP and that the absence of SPIN90 affected APP motility through axons. In particular, internal accumulation of APP in axons was significantly reduced in the absence of SPIN90, suggesting that, in accordance with its modulation of local recycling, SPIN90 is a regulator of APP motility via axons, which is related to its accumulation. Internal accumulation of APP in axons suggests that these may be prime sites of Aβ production. However, it is unclear whether axons or dendrites are more essential for Aβ production. Studies are needed to identify the intracellular fractions, such as early endosomes, recycling endosomes, lysosomes, and autophagosomes, that contain internal APP and the internal organelles associated with to amyloidogenesis. 

Synaptic failure or synaptopathy was shown to be an invariant indicator of AD at an early stage. Synaptic transmission was strongly suppressed in 5xFAD neurons, likely because of the high levels of Aβ production and deposition. However, SPIN90 deficiency ameliorated synaptic transmission and restored age-dependent synaptic depression, perhaps because the reduced level of Aβ production following the depletion of SPIN90 would lead to recovery of synaptic function. Collectively, these findings showed that modulation of APP trafficking altered APP processing and Aβ production, thereby altering synaptic function.

## 4. Materials and Methods

### 4.1. Animals

All animal procedures were approved by the Animal Care and Ethics Committees of the GIST (GIST-2019-096, GIST-2022-001). All experiments with animals were performed using age-matched littermates. The 5xFAD mice, with five familial AD mutations that were introduced in APP695 and PS1 cDNAs (APP K607N/M671L + I716V + V717I and PS1 M146L + L286V), were purchased from Jackson Laboratory (#034840-JAX). SPIN90 KO mice were generated as previously described [39]. Male 5xFAD transgenic mice were backcrossed onto homozygous female SPIN90 KO mice to yield male and female 5xFAD-SPIN90 heterozygous (F1) mice, which were mated to generate four types of F2 mice, WT-SPIN90 WT, WT-SPIN90 KO, 5xFAD-SPIN90 WT, and 5xFAD-SPIN90 KO.

### 4.2. Antibodies and Reagents

Primary antibodies included anti-Aβ 4G8 (SIG-39220, BioLegend, San Diego, CA, USA), anti-SPIN90 (generated in our laboratory), anti-Rab11 (ab3612, Abcam, Cambridge, UK), anti-GFP (sc-9996, Santa Cruz Biotechnology, Dallas, TX, USA), anti-APP (MAB348, EMD Millipore, Darmstadt, Germany), anti-BACE1 (B0681, Sigma, St. Louis, MO, USA), anti-Nicastrin (5665, Cell Signaling Technology, Danvers, MA, USA), anti-β-actin (sc-47778, Santa Cruz Biotechnology, Dallas, TX, USA), anti-p35/p25 (2680, Cell Signaling Technology, MA, USA) and anti-tubulin (T6199, Sigma, St. Louis, MO, USA). Secondary antibodies included horseradish peroxidase (HRP)-conjugated donkey anti-mouse (115-035-006) and anti-rabbit antibodies (111-035-006, Jackson Laboratory, Bar Harbor, ME, USA). Unless otherwise noted, all chemicals were purchased from Sigma.

### 4.3. Plasmids and Transfection

Human APP695 cDNA was generated from a mouse cDNA library of 5xFAD primary neurons that express human APP695 mutant (K607N/M671L + I716V + V717I) and cloned by PCR into pHluorin vector with signal sequences. SPIN90 variants were sub-cloned into pEGFP-C1 and RFP vectors [24]. To generate Rab11 plasmids, human Rab11 cDNA was cloned from a human cDNA library, Q70L and S25N mutants were generated using a QuikChange site-directed mutagenesis kit (200518, Agilent, San Diego, CA, USA), and these Rab11 variants were sub-cloned into the mCherry vector pGEX4T-1. Plasmids were transfected into HEK 293T and HeLa cells using Lipofectamine 3000 supplemented with P3000 reagents and into primary cultured neurons using the Ca^2+^ phosphate precipitation method. In the latter method, plasmids were incubated with 2x HEBS (273 mM NaCl, 9.5 mM KCl, 1.4 mM Na_2_HPO_4_·7H_2_O, 15 mM D-glucose, 42 mM HEPES pH 7.10) containing 2 mM Ca^2+^, after which the mixture was applied to 7–8 DIV neurons as described [30].

### 4.4. Immunohistochemistry and Immunocytochemistry

Brain tissues from male mice were fixed in 10% neutral buffered formalin, embedded in paraffin, and sectioned into 6 μm slices. These samples were deparaffinized in Histoclear (National Diagnostics, Atlanta, GA USA, HS-200), hydrated with ethanol, immunostained with anti-amyloid β (4G8) antibodies and visualized using the Dako REAL EnVision detection system (K5007, Dako, Glostrup, Denmark). Images were acquired using Aperio Image Scanning Scope (Leica Biosystems, Wetzlar, Germany). To identify Aβ aggregates, deparaffinized and hydrated slides were incubated with 1% Thioflavin S (Sigma) for 10 min at RT, washed once each for 3 min with 80% and 95% ethanol, and mounted onto a coverslip with FluoroshieldTM (Sigma) solution. Fluorescence signals were scanned using VS200 Research Slide Scanner (Olympus, Tokyo, Japan). 

For presynaptic terminal imaging, mouse hippocampal neurons were transfected with corresponding pairs of constructs (VAMP2-mCh, GFP-SPIN90, mCh-Rab11 and pH-APP695) 8 days after plating. The neurons were fixed with 4% paraformaldehyde (PFA) 14–16 days after plating. Images were acquired using PL APO 63x (1.32NA) or PL Fluor 40x (1.0 N.A.) objectives of a Leica DMRBE microscope along with a CoolSNAP HQ camera (Photometrics, Tocson, AZ, USA) driven by MetaMorph software(ver. 6.1), as previously described [30].

### 4.5. Cell Culture and Primary Neuron Culture

SPIN90 knock-downed (KD) HeLa cells were generated using the Mission RNAi system (SHCLNV, Sigma), as described [23]. HeLa control/KD and HEK 293T cells were grown in Dulbecco’s modified Eagle’s Media (DMEM; Gibco, Carlsbad, CA, USA) supplemented with 10% (*v*/*v*) fetal bovine serum (Hyclone, Logan, UT, USA), 100 units/100 mg/mL penicillin-streptomycin (Gibco, Waltham, MA, USA), and 2 μg/mL puromycin to select for SPIN90 KD cells. To culture primary neurons, cortical or hippocampal regions were dissected from postnatal day 1–3 Sprague Dawley rats (DBL, Eumseong-gun, Chungcheongbuk-do, Republic of Korea) or C57/J mice, dissociated, and plated onto poly-ornithine-coated coverslips. The cells were transfected 8 days later and further incubated in culture media as previously described [30]. 

### 4.6. Protein-Protein Interaction Assays and Western Blotting

Duolink proximity ligation assays (PLA) were performed using Duolink kits (DUO92007, Sigma), according to the manufacturer’s instructions. Cells were fixed with 4% PFA, permeabilized with 0.5% Triton X-100, and incubated with mouse anti-SPIN90 and rabbit anti-Rab11 primary antibodies. The cells were subsequently incubated with secondary PLA probes, which contain a synthetic oligonucleotide that hybridized with and ligated two adjacent molecules separated by less than ~40 nm. Ligation resulted in the amplification of fluorescence signals, which were detected with a FV1000 confocal microscope (Olympus). For glutathione s-transferase (GST) pull-down assays, GST tagged recombinant proteins were purified from BL21 bacteria and immobilized on glutathione-Sepharose beads (Incospharm, Daejeon, Korea) in buffer A (20 mM Tris-HCl pH 8.0, 1 mM EGTA, 150 mM NaCl, 0.5% sodium deoxycholate, 1 mM phenylmethylsulfonyl fluoride (PMSF). Cells expressing GFP-SPIN90 were lysed in pull-down assay buffer B (25 mM HEPES, 100 mM NaCl, 5 mM MgCl2, 1% Nonidet P40 (NP-40), 10% glycerol, 1 mM PMSF, protease inhibitor cocktail), and the lysates were incubated with purified GST conjugating proteins. After incubation for 2 h at 4 °C, beads bound with proteins were washed four times with buffer B. The concentrations of cell lysates were measured using PierceTM bicinchoninic acid (BCA) protein assay kits (23225, Thermo Fisher Scientific, Waltham, MA, USA). The lysates were resolved on sodium dodecyl sulfate polyacrylamide gel electrophoresis (SDS-PAGE) gels, and the proteins transferred to polyvinylidene difluoride (PVDF) membranes. The membranes were blocked by incubation with buffer containing 5% bovine serum albumin or skim milk and incubated with primary antibodies at 4 °C overnight. After washing, the membranes were incubated with the appropriate HRP-conjugated secondary antibodies. Antibody binding was detected by WesternBrightTM enhanced chemiluminescence (ECL) (Advansta, San Jose, CA, USA) using a LAS-2000 (Fujifilm, Tokyo, Japan).

### 4.7. BACE1 Activity Assay

BACE1 activity was measured using β-secretase activity fluorometric assay kits (MAK237, Sigma) according to the manufacturer’s instructions. Briefly, tissues were homogenized and sonicated with β-secretase extraction buffer. Tissue lysates were incubated with fluorogenic β-secretase substrate, containing a β-cleavage site. Fluorescence was measured at Ex/Em = 33–355 nm/495–510 nm using Flexstation 3 (Molecular Devices, San Jose, CA, USA).

### 4.8. Live-Cell Imaging for Synapse Physiology

Presynaptic terminal live imaging of synaptic transmission and APP695 recycling were evaluated by transfecting constructs (vG-pH and pH-APP695) 6–8 days after plating, with experiments performed 10–21 days after plating. Coverslips were mounted in a stimulation chamber with laminal-flow perfusion on the stage of a custom-built laser-illuminated epifluorescence microscope. Live images were acquired with an Andor iXon Ultra 897 (Model #DU-897U-CS0-#BV; Belfast, Northern Ireland) back-illuminated EM CCD camera. The light source was a diode-pumped OBIS 488 laser (Coherent, Santa Clara, CA, USA), with shutters activated by synchronizing the TTL on/off signal from the EMCCD camera during acquisition. The fluorescence excitation wavelength was 498 nm, using dichroic filters (Chroma, Irvine, CA, USA) for pHluorin, and the emission wavelengths were 500–550 nm. Images were acquired using a 40× (1.3 NA) Fluar Zeiss objective lens. Action potentials (AP) were evoked by passing a current pulse of 1 ms through platinum-iridium electrodes from an isolated current stimulator (World Precision Instruments, Hitchin, Hertfordshire, UK). Neurons were perfused with Tyrode’s buffer, containing 119 mM NaCl, 2.5 mM KCl, 2 mM CaCl2, 2 mM MgCl2, 25 mM HEPES, 30 mM glucose, 10 μM 6-cyano-7-nitroquinoxaline-2,3-dione (CNQX), and 50 μM D,L-2-amino-5-phosphonovaleric acid (AP5), adjusted to pH 7.4. All experiments were performed at 30 °C. Images for vG-pH-transfected neurons were stimulated for 10 s at 10 Hz. NH_4_Cl was applied to measure the size of the total synaptic vesicle pool. Images were acquired at 2 Hz with 50 ms exposure, as described previously [40]. To image APP695 distribution, MES-buffered Tyrode’s solution (pH 5.5) was initially applied to pH-APP695 transfected neurons; the neurons were washed with normal Tyrode’s buffer, followed by the application of NH_4_Cl (pH 7.4) buffer, as described previously [30]. To evaluate APP695 recycling, neurons transfected with pH-APP695 were stimulated with 20 Hz for 30 s.

### 4.9. Image Analysis

All images were analyzed using Image J software (Image J. Available online: http://rsb.info.nih.gov/ij, accessed on 30 July 2022). To analyze IHC data, the relative areas of Aβ deposition were quantified. Because signals representing areas of Aβ deposition partially overlapped or were connected in samples from older mice, ambiguity was encountered in determining the number of Aβ deposits. Binding of antibodies to brain sections was measured by calculating the percent total areas of the hippocampus and subiculum positive for Aβ signals on each of three slides and determining their averages. Because the fluorescence signals of thioflavin S were unambiguous, the numbers of Aβ aggregates were counted by determining the numbers of thioflavin S positive puncta with areas greater than six pixels (72 μm^2^). SPIN90-Rab11 interaction signals in the Duolink assay were quantified by counting the number of PLA positive puncta in each cell. pHluorin signals were analyzed using the ImageJ plugin time-series analyzer with minor modifications. For synaptic transmission and APP695 recycling, vG-pH-positive boutons (over 50 boutons per neuron) and pH-APP695, respectively, were selected as regions of interest (>1.5-μm diameter). Fluorescence traces were analyzed using Origin Pro 2020. The pHluorin signal amplitudes (ΔF values of each 100 AP response or surface distribution of pH-APP695) were normalized relative to the maximum value of NH_4_Cl, as previously described [30]. Accumulations of pH-APP695 were defined as puncta >120 pixels in size, as determined using ImageJ software.

### 4.10. Statistics

OriginPro (ver. 2020) was utilized for statistical analysis. All data are presented as mean ± standard error of the mean (SEM). Data in two groups were compared by Student’s *t*-tests, whereas data in three or more groups were compared by one-way ANOVA. A *p*-value < 0.05 was defined as statistically significant.

## 5. Conclusions

APP trafficking is an important process in the generation of Aβ. The present study showed that SPIN90 regulates APP processing by modulating APP trafficking. Along with Rab11, SPIN90 regulates the distribution of APP in axons and dendrites, and of activity-dependent local recycling. Depletion of SPIN90 reduced surface APP on axons while increasing surface APP on dendrites, while strongly enhancing activity-driven recycling of APP. This study also found that APP axonal trafficking and intracellular accumulation were related to SPIN90. APP moves along with SPIN90 via axons, with less APP accumulating in endosomes in the absence of SPIN90. Over time, SPIN90 deficiency can influence Aβ production and deposition and restore synaptic functionality. 

## Figures and Tables

**Figure 1 ijms-23-10563-f001:**
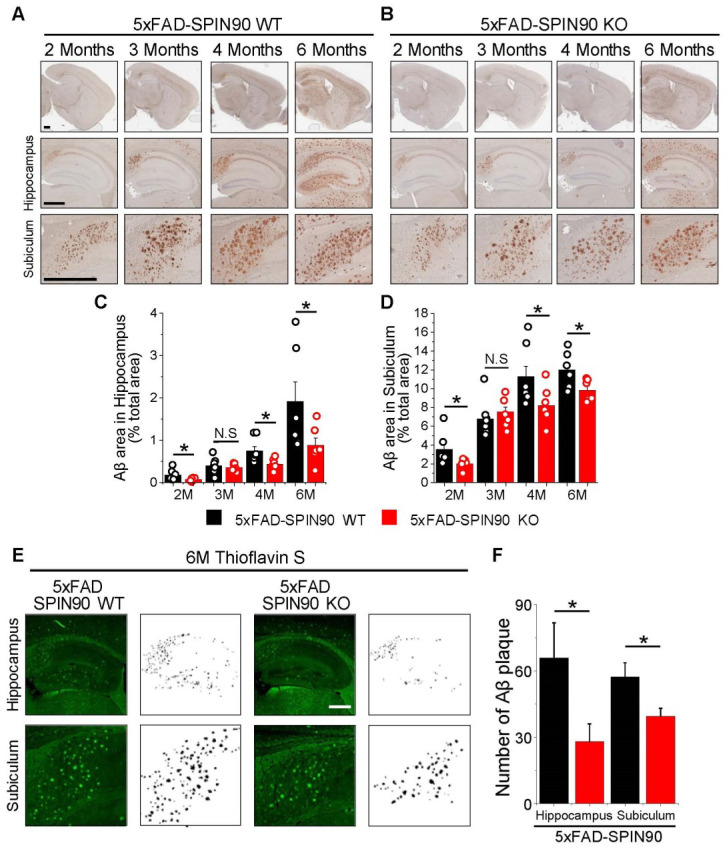
SPIN90 knockout reduces amyloid β (Aβ) deposition in AD model mice. (**A**,**B**) Representative images of Aβ plaques in the brains of 5xFAD-SPIN90 WT (**A**) and 5xFAD-SPIN90 KO mice (**B**), as shown by Dab reaction (brown) and hematoxylin straining (blue). Brains of mice aged 2, 3, 4, and 6 months were fixed and stained with anti-Aβ antibody. (Top) The entire area of a hemisphere; (middle) magnified images of the hippocampus; (bottom) magnified images of the subiculum. Scale bars, 500 μm. (**C**,**D**) Mean areas of Aβ deposition in the hippocampus (**C**) and subiculum (**D**) of brains from 2-, 3-, 4-, and 6-month old 5xFAD-SPIN90 WT (Black) and 5xFAD-SPIN90 KO (Red) mice. Total areas of Aβ plaque were quantified and normalized relative to areas of the hippocampus or subiculum. Hippocampus: [2M; Aβ plaque]_5xFAD-SPIN90WT_ = 0.18 ± 0.05%, n = 18 (6 mice), [2M; Aβ plaque]_5xFAD-SPIN90KO_ = 0.08 ± 0.01%, n = 18 (6 mice), [3M; Aβ plaque]_5xFAD-SPIN90WT_ = 0.4 ± 0.07%, n = 21 (7 mice), [3M; Aβ plaque]_5xFAD-SPIN90KO_ = 0.36 ± 0.03%, n = 21 (7 mice), [4M; Aβ plaque]_5xFAD-SPIN90WT_ = 0.75 ± 0.1%, n = 21 (7 mice), [4M; Aβ plaque]_5xFAD-SPIN90KO_ = 0.44 ± 0.04%, n = 21 (7 mice), [6M; Aβ plaque]_5xFAD-SPIN90WT_ = 1.9 ± 0.47%, n = 18 (6 mice), [6M; Aβ plaque]_5xFAD-SPIN90KO_ = 0.89 ± 0.17, n = 18 (6 mice). Subiculum: [2M; Aβ plaque]_5xFAD-SPIN90WT_ = 3.58 ± 0.66, n = 18 (6 mice), [2M; Aβ plaque]_5xFAD-SPIN90KO_ = 2.04 ± 0.2, n = 18 (6 mice), [3M; Aβ plaque]_5xFAD-SPIN90WT_ = 6.79 ± 0.7, n = 21 (7 mice), [3M; Aβ plaque]_5xFAD-SPIN90KO_ = 7.54 ± 0.5, n = 21 (7 mice), [4M; Aβ plaque]_5xFAD-SPIN90WT_ = 11.29 ± 1.09, n = 21 (7 mice), [4M; Aβ plaque]_5xFAD-SPIN90KO_ = 8.22 ± 0.67, n = 21 (7 mice), [6M; Aβ plaque]_5xFAD-SPIN90WT_ = 12.02 ± 0.73, n = 18 (6 mice), [6M; Aβ plaque]_5xFAD-SPIN90KO_ = 9.85 ± 0.43, n = 18 (6 mice). (**E**) Representative images of thioflavin S- stained Aβ plaques in the hippocampus and subiculum in the brains of 6-month-old 5xFAD-SPIN90 WT (left) and 5xFAD-SPIN90 KO (right) mice. Images are shown inverted to clearly display Aβ plaques. Scale bars, 500 μm. (**F**) Quantification of Aβ aggregates in the hippocampus and subiculum of 5xFAD-SPIN90 WT and 5xFAD-SPIN90 KO mice. [Hippocampus: Aβ aggregates]_5xFAD-SPIN90WT_ = 65.83 ± 15.88 (n = 6), [Hippocampus: Aβ aggregates]_5xFAD-SPIN90KO_ = 28.17 ± 7.94 (n = 6), [Subiculum: Aβ aggregates]_5xFAD-SPIN90WT_ = 57.33 ± 6.32 (n = 6), [Subiculum: Aβ aggregates]_5xFAD-SPIN90KO_ = 39.5 ± 3.61 (n = 6). * *p* < 0.05. N.S.; not significant.

**Figure 2 ijms-23-10563-f002:**
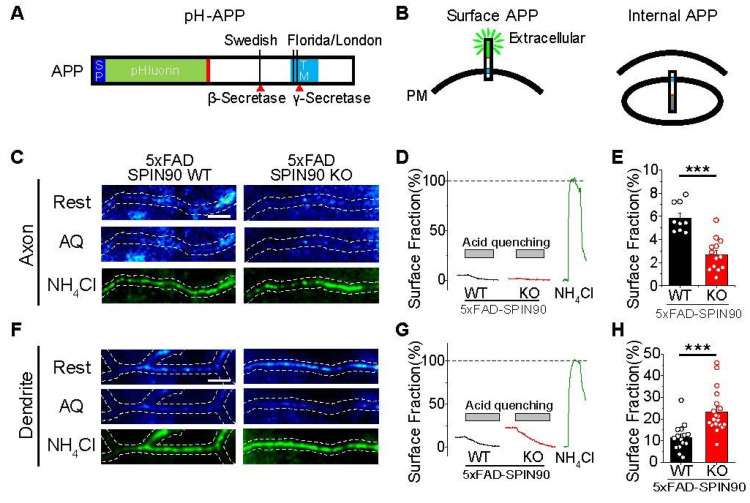
Deficiency of SPIN90 alters surface fraction of APP in axons and dendrites of 5xFAD neurons. (**A**) Schematic diagram of pHluorin-conjugated APP (pH-APP). (**B**) Illustration of surface pH-APP (left) and endosomal pH-APP (right). (**C**) Representative images of pH-APP in axons of 5xFAD-SPIN90 WT (left) and 5xFAD-SPIN90 KO (right) neurons at rest (top), after acid quenching (middle), and after NH_4_Cl treatment (bottom). Scale bar: 5 μm. (**D**) Representative traces of pH-APP in response to acid quenching in axons of 5xFAD-SPIN90 WT (black) and 5xFAD-SPIN90 KO (red) neurons. (**E**) Mean surface fraction of APP in axons of 5xFAD-SPIN90 WT and 5xFAD-SPIN90 KO neurons. [Surface fraction: axon]_5xFAD-SPIN90WT_ = 5.85 ± 0.42% (n = 9 cells), [Surface fraction: axon]_5xFAD-SPIN90KO_ = 2.66 ± 0.38% (n = 13 cells). (**F**) Representative images of pH-APP in dendrites of 5xFAD-SPIN90 WT (left) and 5xFAD-SPIN90 KO (right) neurons at rest (top), after acid quenching (middle), and after NH_4_Cl treatment (bottom). Scale bar, 5 μm. (**G**) Representative traces of pH-APP in response to acid quenching in dendrites of 5xFAD-SPIN90 WT (black) and 5xFAD-SPIN90 KO (red) neurons. (**H**) Mean surface fraction of APP in dendrites of 5xFAD-SPIN90 WT and 5xFAD-SPIN90 KO neurons. [surface fraction: dendrite]_5xFAD-SPIN90WT_ = 11.56 ± 1.63% (n = 15 cells), [surface fraction: dendrite]_5xFAD-SPIN90KO_ = 23.26 ± 2.23% (n = 19 cells). *** *p* < 0.001.

**Figure 3 ijms-23-10563-f003:**
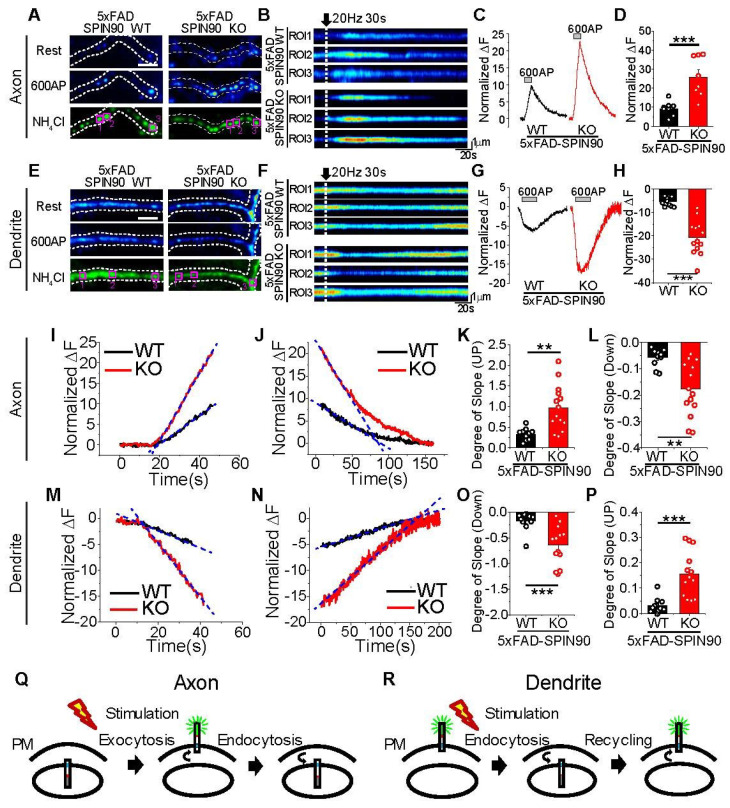
Activity-driven recycling of APP in axons and dendrites is enhanced in 5xFAD-SPIN90 KO neurons. (**A**,**E**) Representative images of pH-APP in axons (**A**) and dendrites (**E**) of 5xFAD-SPIN90 WT (left) and 5xFAD-SPIN90 KO (right) neurons at rest (top), in the presence of an action potential (AP) of 600 (middle), and after NH_4_Cl treatment (bottom). Scale bar: 5 μm. (**B**,**F**) Kymographs of pH-APP in corresponding areas of axons (**B**) and dendrites (**F**) in 5xFAD-SPIN90 WT (top) and 5xFAD-SPIN90 KO (bottom) neurons in response to 600AP. (**C**,**G**) Representative traces of pH-APP in axons (**C**) and dendrites (**G**) of 5xFAD-SPIN90 WT (black) and 5xFAD-SPIN90 KO (red) neurons in response to 600AP. (**D**,**H**) Mean peak amplitudes in axons (**D**) and dendrites (**H**) of 5xFAD-SPIN90 WT and 5xFAD-SPIN90 KO neurons in response to 600AP. [600AP: axon]_5xFAD-SPIN90WT_ = 9.30 ± 1.31% (n = 8 cells), [600AP: axon]_5xFAD-SPIN90KO_ = 25.87 ± 3.69% (n = 8 cells); [600AP: dendrite]_5xFAD-SPIN90WT_ = −5.48 ± 0.62% (n = 10 cells), [600AP: dendrite]_5xFAD-SPIN90KO_ = −20.80 ± 2.05% (n = 13 cells). Representative traces of pH-APP with linear fit-line during stimulation of (**I**) axons and (**M**) dendrites and after stimulation of (**J**) axons and (**N**) dendrites in 5xFAD-SPIN90 WT (black) and 5xFAD-SPIN90 KO (red) neurons. (**K**,**O**) Mean slopes as determined by linear fitting during stimulation of axons (**K**) and dendrites (**O**) in 5xFAD-SPIN90 WT and 5xFAD-SPIN90 KO neurons. [Degree of slope: axon]_5xFAD-SPIN90WT_ = 0.33 ± 0.05 (n = 10 cells); [Degree of slope: axon]_5xFAD-SPIN90KO_ = 0.97 ± 0.15 (n = 14 cells); [Degree of slope: dendrite]_5xFAD-SPIN90WT_ = −0.17 ± 0.05 (n = 11 cells); [Degree of slope: dendrite]_5xFAD-SPIN90KO_ = −0.65 ± 0.11 (n = 13 cells). (**L**,**P**) Mean slopes as determined by linear fitting after stimulation of axons (**L**) and dendrites (**P**) in 5xFAD-SPIN90 WT and 5xFAD-SPIN90 KO neurons. [Degree of slope: axon]_5xFAD-SPIN90WT_ = −0.057 ± 0.01 (n = 10 cells); [Degree of slope: axon]_5xFAD-SPIN90KO_ = −0.18 ± 0.03 (n = 14 cells); [Degree of slope: dendrite]_5xFAD-SPIN90WT_ = 0.032 ± 0.009 (n = 11 cells); [Degree of slope: dendrite]_5xFAD-SPIN90KO_ = 0.16 ± 0.03 (n = 13 cells). (**Q**) Schematic illustration of activity-driven pH-APP recycling in axons (endosome-surface-endosome). (**R**) Schematic illustration of activity-driven pH-APP recycling in dendrites (surface-endosome-surface). ** *p* < 0.01, *** *p* < 0.001.

**Figure 4 ijms-23-10563-f004:**
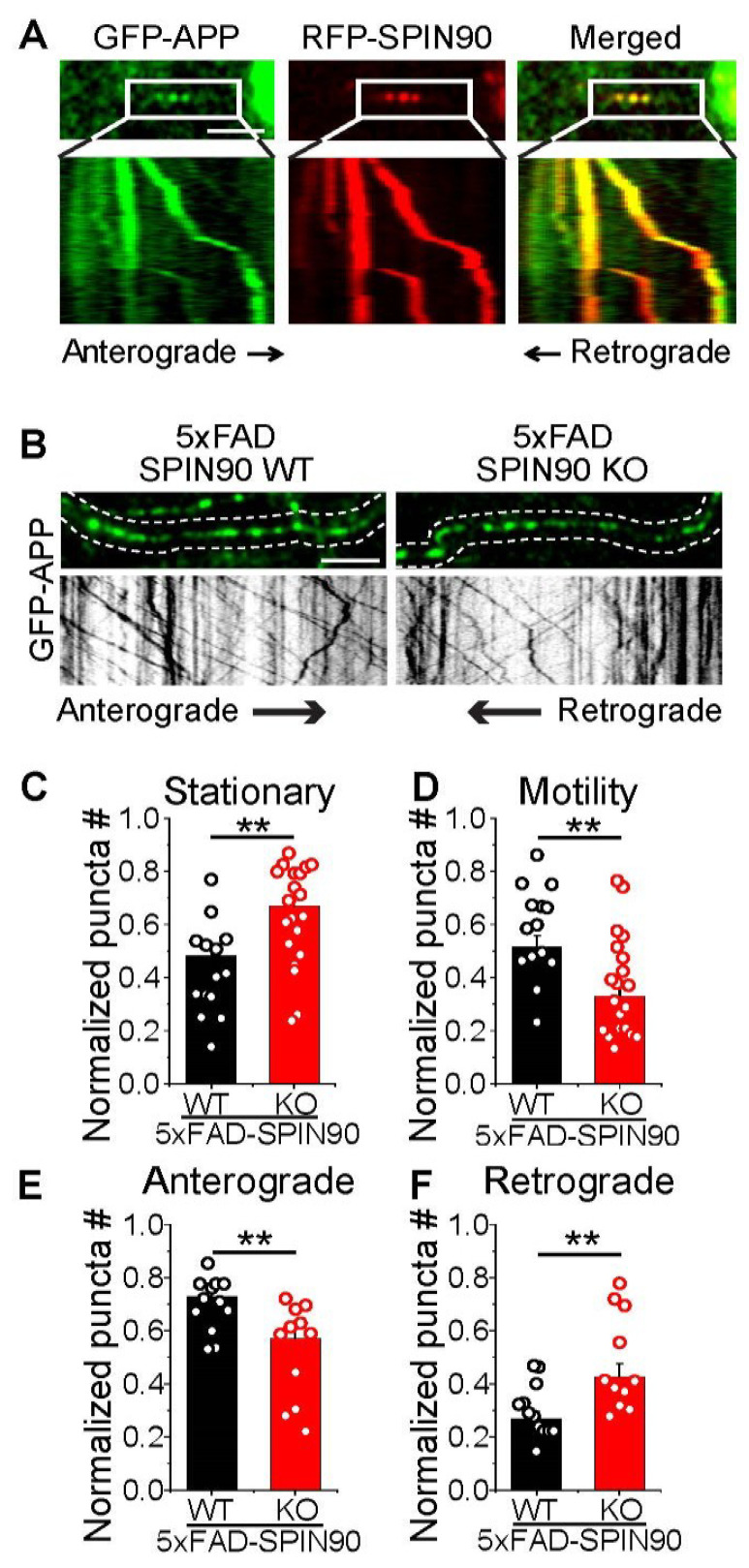
SPIN90 modulates axonal motility of APP. (**A**) Representative images of GFP-APP and RFP-SPIN90 in rat hippocampal neurons. Selected images from the boxed area in top and kymographs (bottom) of GFP-APP and RFP-SPIN90. Scale bar: 5 μm. (**B**) (Top) Representative images of GFP-APP in 5xFAD-SPIN90 WT (left) and 5xFAD-SPIN90 KO (right) neurons. (Bottom) Kymographs of GFP-APP in 5xFAD-SPIN90 WT (left) and 5xFAD-SPIN90 KO (right) neurons. Scale bar: 10 μm. (**C**) Mean stationary APP in 5xFAD-SPIN90 WT and 5xFAD-SPIN90 KO neurons. [stationary APP]_5xFAD-SPIN90WT_ = 0.48 ± 0.04 (n = 14 cells); [stationary APP]_5xFAD-SPIN90KO_ = 0.67 ± 0.04 (n = 20 cells). (**D**) Mean motile APP in 5xFAD-SPIN90 WT and 5xFAD-SPIN90 KO neurons. [motility APP]_5xFAD-SPIN90WT_ = 0.52 ± 0.04 (n = 14 cells); [motility APP]_5xFAD-SPIN90KO_ = 0.33 ± 0.04 (n = 20 cells). (**E**) Mean anterograde transport of APP in 5xFAD-SPIN90 WT and 5xFAD-SPIN90 KO neurons. [anterograde transport of APP]_5xFAD-SPIN90WT_ = 0.73 ± 0.03 (n = 12 cells); [anterograde transport of APP]_5xFAD-SPIN90KO_ = 0.57 ± 0.05 (n = 11 cells). (**F**) Mean retrograde transport of APP in 5xFAD-SPIN90 WT and 5xFAD-SPIN90 KO neurons. [retrograde transport of APP]_5xFAD-SPIN90WT_ = 0.27 ± 0.03 (n = 12 cells); [retrograde transport of APP]_5xFAD-SPIN90KO_ = 0.43 ± 0.05 (n = 11 cells). ** *p* < 0.01. #; number.

**Figure 5 ijms-23-10563-f005:**
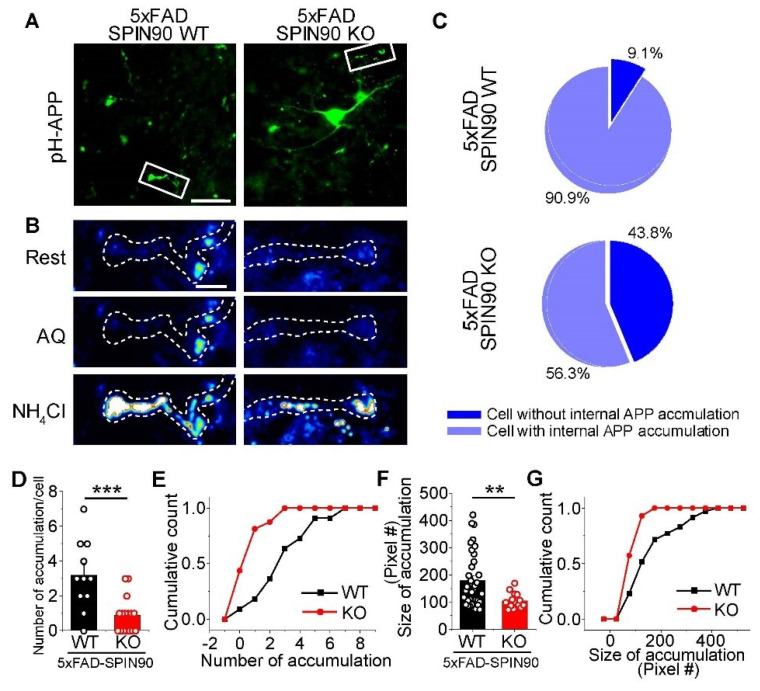
Depletion of SPIN90 regulates internal accumulation of APP in 5xFAD neurons. (**A**) Representative images of pH-APP accumulation in 5xFAD-SPIN90 WT (left) and 5xFAD-SPIN90 KO (right) neurons. Cells transfected with pH-APP were fixed at DIV 14~17. Scale bar: 20 μm. (**B**) Magnified images of the boxed area in (**A**) at rest (top), during acid quenching (middle), and during NH_4_Cl application (bottom). Scale bar: 5 μm. (**C**) Pie chart showing the percentages of cells with/without APP endosomal accumulation. [Cell without APP endosomal accumulation]_5xFAD-SPIN90WT_ = 9.09% (n = 11 cells); [Cell without APP endosomal accumulation]_5xFAD-SPIN90KO_ = 43.75% (n = 16 cells); [Cell with APP endosomal accumulation]_5xFAD-SPIN90WT_ = 90.91% (n = 11 cells); [Cell with APP endosomal accumulation]_5xFAD-SPIN90KO_ = 56.25% (n = 16 cells). (**D**) Mean pH-APP accumulation in 5xFAD-SPIN90 WT and 5xFAD-SPIN90 KO neurons. [pH-APP accumulation/Cell]_5xFAD-SPIN90WT_ = 3.18 ± 0.6 (n = 11 cells); [pH-APP accumulation/Cell]_5xFAD-SPIN90KO_ = 0.88 ± 0.26 (n = 16 cells). (**E**) Cumulative pH-APP accumulation in 5xFAD-SPIN90 WT (black) and 5xFAD-SPIN90 KO (red) neurons. (**F**) Mean sizes of endosomal pH-APP accumulation in 5xFAD-SPIN90 WT and 5xFAD-SPIN90 KO neurons. [size of pH-APP accumulation]_5xFAD-SPIN90WT_ = 179.34 ± 17.03 pixel (n = 35 puncta); [size of pH-APP accumulation]_5xFAD-SPIN90KO_ = 105 ± 7.63 pixel (n = 14 puncta). (**G**) Cumulative sizes of endosomal pH-APP accumulation in 5xFAD-SPIN90 WT (black) and 5xFAD-SPIN90 KO (red) neurons. ** *p* < 0.01, *** *p* < 0.001.

**Figure 6 ijms-23-10563-f006:**
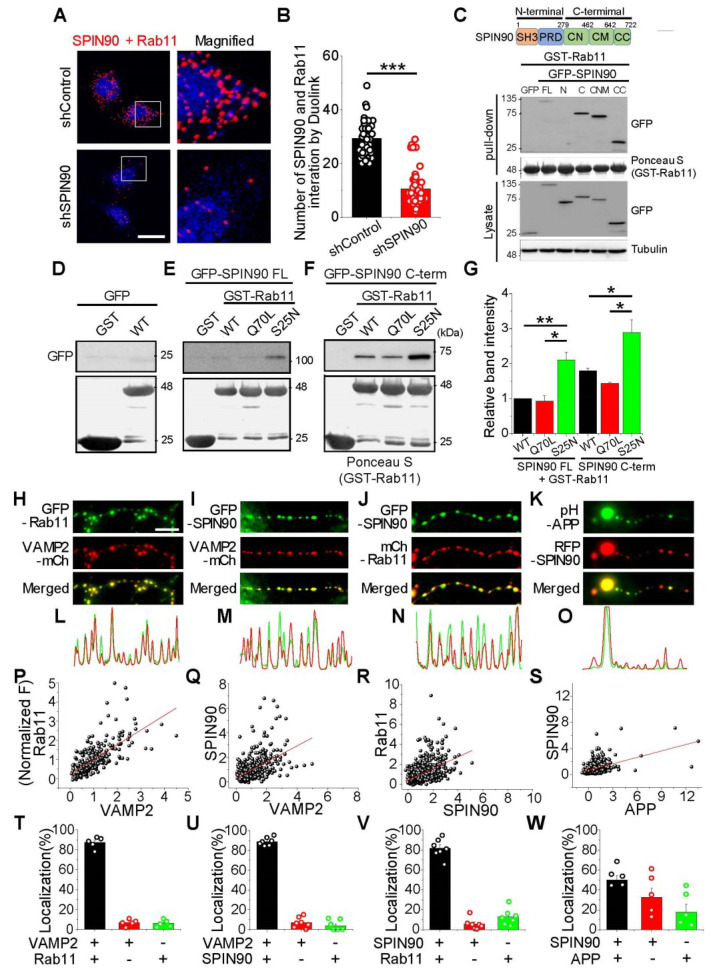
SPIN90 interacts with inactive Rab11. (**A**) Left: Representative images of SPIN90 and Rab11 interaction by Duolink in HeLa cells bearing shControl (top) and shSPIN90 (bottom). Right: Magnified image of the box areas. Red puncta indicate in situ interactions between SPIN90 and Rab11. (**B**) Mean numbers of puncta in shControl and shSPIN90-HeLa cells. [Puncta #]_shControl_ = 20.25 ± 0.05 (564 cells, n = 4 experiments), [Puncta #]_shSPIN90_ = 9.92 ± 0.58 (272 cells, n = 4 experiments). scale bar: 20 μm. (**C**) Representative immunoblot of SPIN90 and Rab11 interactions, as shown by GST-binding assays. Cells transfected with GFP-SPIN90 full-length (FL), N-term (N), C-term (C), CN+CM (CNM), and CC domain were lysed and the lysates subsequently incubated with GST-Rab11 protein. GST-Rab11 was pull-downed and interactions were verified by immunoblotting with anti-GFP antibody. (**D**–**F**) Representative immunoblots of the interactions of SPIN90 and Rab11. Interactions of the active/inactive forms of Rab11 with full-length SPIN90 (**E**), or with its C-terminus (**F**). (**G**) Quantification of relative band intensities in (**E**,**F**). The intensities, normalized to the band intensities of GST-Rab11 WT and SPIN90 FL. [GST-Rab11 Q70L]_SPIN90 FL_ = 0.93 ± 0.16; [GST-Rab11 S25N]_SPIN90 FL_ = 2.11 ± 0.22; [GST-Rab11 WT]_SPIN90 C-term_ = 1.8 ± 0.07; [GST-Rab11 Q70L]_SPIN90 C-term_ = 1.43 ± 035; [GST-Rab11 S25N]_SPIN90 C-term_ = 2.89 ± 0.36 (n = 3). (**H**–**K**) Representative images of Rab11, SPIN90, APP, and VAMP2 in hippocampal neurons. Neurons were cotransfected with GFP-Rab11/VAMP2-mCh (**H**), GFP-SPIN90/VAMP2-mCh (**I**), GFP-SPIN90/mCh-Rab11 (**J**) and pH-APP/RFP-SPIN90 (**K**) at DIV 8 and fixed at DIV 14–16. (**L**–**O**) Line scans of each image. (**P**–**S**) Distribution and correlation of expression levels of GFP-Rab11/VAMP2-mCh (**P**: n = 277 synapses), GFP-SPIN90/VAMP2-mCh (**Q**: n = 470 synapses), GFP-SPIN90/mCh-Rab11 (**R**: n = 887 synapses) and pH-APP/RFP-SPIN90 (**S**: n = 330 synapses). (**T**–**W**) Quantification of colocalization of corresponding proteins (**T**: GFP-Rab11/VAMP2-mCh, **U**: GFP-SPIN90/VAMP2-mCh, **V**: GFP-SPIN90/mCh-Rab11, and **W**: pH-APP/RFP-SPIN90). * *p* < 0.05, ** *p* < 0.01 *** *p* < 0.001.

**Figure 7 ijms-23-10563-f007:**
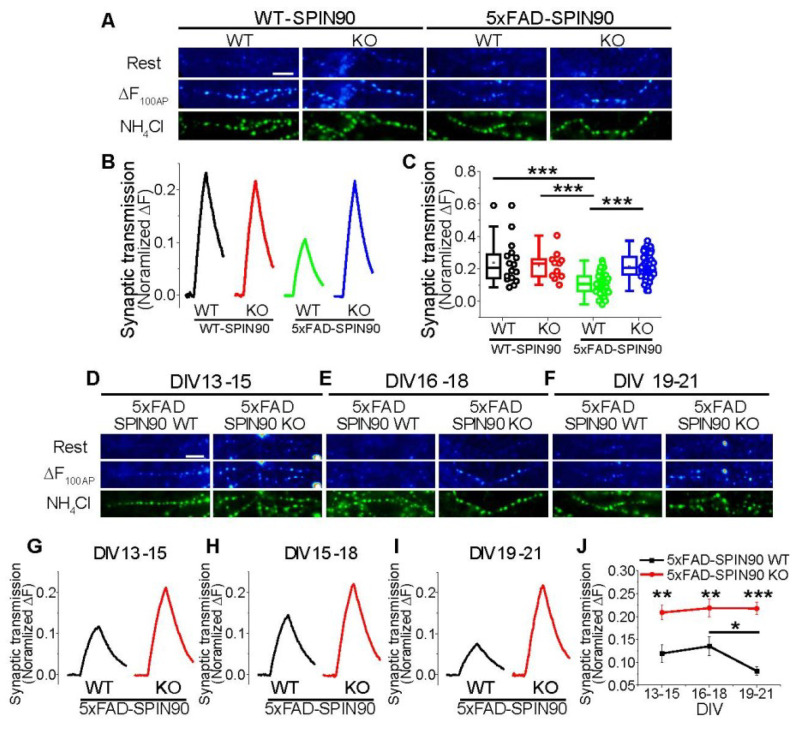
SPIN90 deficiency restores synaptic function in 5xFAD neurons. (**A**) Representative synapse images of vG-pH at rest (top) and differences from 100 AP-treated (ΔF_100AP_, middle), and NH_4_Cl-treated (bottom) neurons from WT-SPIN90 WT, WT-SPIN90 KO, 5xFAD-SPIN90 WT and 5xFAD-SPIN90 KO mice. Scale bar: 5 μm. (**B**) Representative traces of vG-pH in response to 100AP in WT-SPIN90 WT (black), WT-SPIN90 KO (red), 5xFAD-SPIN90 WT (green) and 5xFAD-SPIN90 KO (blue) neurons. (**C**) Mean synaptic transmission in WT-SPIN90 WT, WT-SPIN90 KO, 5xFAD-SPIN90 WT and 5xFAD-SPIN90 KO neurons. [synaptic transmission]_WT-SPIN90WT_ = 0.24 ± 0.03 (n = 16 cells); [synaptic transmission]_WT-SPIN90KO_ = 0.22 ± 0.026 (n = 11 cells); [synaptic transmission]_5xFAD-SPIN90WT_ = 0.11 ± 0.01 (n = 46 cells); [synaptic transmission]_5xFAD-SPIN90KO_ = 0.21 ± 0.01 (n = 63 cells). (**D**–**F**) Representative synapse images of vG-pH at rest (top) and differences from 100 AP treated (ΔF_100AP_, middle), and NH_4_Cl treated (bottom) neurons from 5xFAD-SPIN90 WT and 5xFAD-SPIN90 KO mice at DIVs (**D**) 13–15, (**E**) 16–18, and (**F**) 19–21. Scale bar: 5 μm. (**G**–**I**) Representative traces of vG-pH in response to 100AP in 5xFAD-SPIN90 WT (black) and 5xFAD-SPIN90 KO (red) neurons at DIVs (**G**) 13–15, (**H**) 16–18, and (**I**) 19–21. (**J**) The mean synaptic transmission in 5xFAD-SPIN90 WT (black) and 5xFAD-SPIN90 KO (red) neurons in various time window at DIVs 13–15, 16–18 and 19–21. [synaptic transmission: DIV 13–15]_5xFAD-SPIN90WT_ = 0.12 ± 0.02 (n = 15 cells), [synaptic transmission: DIV 16–18]_5xFAD-SPIN90WT_ = 0.13 ± 0.02 (n = 12 cells), [synaptic transmission: DIV 19–21]_5xFAD-SPIN90WT_ = 0.08 ± 0.01 (n = 17 cells), [synaptic transmission: DIV 13–15]_5xFAD-SPIN90KO_ = 0.21 ± 0.02 (n = 22 cells), [synaptic transmission: DIV 16–18]_5xFAD-SPIN90KO_ = 0.22 ± 0.02 (n = 19 cells), [synaptic transmission: DIV 19-21]_5xFAD-SPIN90KO_ = 0.22 ± 0.01 (n = 22 cells). * *p* < 0.05, ** *p* < 0.01, *** *p* < 0.001.

## Data Availability

Not applicable.

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
