# Peer review of "SPIN90 Deficiency Ameliorates Amyloid β Accumulation by Regulating APP Trafficking in AD Model Mice"

_ijms, 2022, doi:10.3390/ijms231810563_

Round 1
Reviewer 1 Report
In this study, Oh et al. showed SPIN90 being regulator of APP trafficking and Aβ accumulation in AD model mice. SPIN90 deficiency was found to decrease Aβ plaques in 5xFAD mice and neurons. SPIN90 was further observed to spatially regulate the APP distribution between axons and dendrites through Rab11+ endosomes via directly interacting with Rab11. Moreover, SPIN90 was found to localize at presynapses and synapses, and its deficiency did restore the synaptic function in 5xFAD neurons.
The study did employ wide-spectrum of methodologies such as IHC, ICC, PLA, coIP, and live-cell imaging, as well as models such as 5xFAD AD model mice, 5xFAD AD model neurons, HeLa and HEK cells; and as overall, it is a quantitatively well-investigated study. However, authors are highly encouraged to consider the following points to improve the study.
* As authors did show that the protein levels of BACE1 and Nicastrin (γ-Secretase complex member) are not the factors on APP processing mediated by SPIN90. Previous studies did reveal that p25/Cdk5-mediated phosphorylation activates BACE1 (PMID: 26317805). Did authors consider the BACE1 activation status?
* Authors did employ two different strategies (Area in Fig.1A-D, count numbers in Fig.1E-F) to measure Aβ plaques. Scientific reasoning is kindly asked and should be clearly mentioned in the text.
* How is the counting methodology for the Aβ plaques? (e.g. what is the lowest threshold size limit of clusters to be considered?). Also, have authors happened to see any difference in size distribution of Aβ plaque clusters between 5xFAD SPIN90wt and 5xFAD SPIN90ko (Fig.1E-F)?
* It has been previously shown that APP has functions of the mediation of post-synaptic density composition and activity (PMID: 19164281). As authors did show in this study that SPIN90 localize at presynapses and synapses and its deficiency regulates synaptic funtions; have authors measured the APP accumulation difference in post-synaptic density between 5xFAD SPIN90wt and 5xFAD SPIN90ko?
* What are the criteria on selection of ROIs?
* Did authors introduce the same 5xFAD mutations of APP to the pH-APP vector? If so, please correctly annotate in the scheme Fig.2A.
* There is no time-lapse live-imaging data on GFP-APP and RFP-SPIN90 move together through axons in Fig.4A, as written in the text (Lines 231-232). Authors are kindly asked to provide the supporting data.
* Did authors test the dynamics of the axonal motility of APP whether it's affected upon RAB11 attenuation or overexpression compared to endogenous level?
* Via what exact mechanism, the depletion of SPIN90 reduces the Aβ plaque level? For example, does SPIN90 sequester BACE1 into APP+ RAB11 endosomes? Did authors check BACE1-APP colocalization difference between 5xFAD SPIN90wt and 5xFAD SPIN90ko?
Author Response
Reviewer #1
In this study, Oh et al. showed SPIN90 being regulator of APP trafficking and Aβ accumulation in AD model mice. SPIN90 deficiency was found to decrease Aβ plaques in 5xFAD mice and neurons. SPIN90 was further observed to spatially regulate the APP distribution between axons and dendrites through Rab11+ endosomes via directly interacting with Rab11. Moreover, SPIN90 was found to localize at presynapses and synapses, and its deficiency did restore the synaptic function in 5xFAD neurons.
The study did employ wide-spectrum of methodologies such as IHC, ICC, PLA, coIP, and live-cell imaging, as well as models such as 5xFAD AD model mice, 5xFAD AD model neurons, HeLa and HEK cells; and as overall, it is a quantitatively well-investigated study. However, authors are highly encouraged to consider the following points to improve the study.
- As authors did show that the protein levels of BACE1 and Nicastrin (γ-Secretase complex member) are not the factors on APP processing mediated by SPIN90. Previous studies did reveal that p25/Cdk5-mediated phosphorylation activates BACE1 (PMID: 26317805). Did authors consider the BACE1 activation status?
Response1-
To verify determine whether BACE1 activity is also altered, we measured p25 expression level, BACE1 activity and the level of the dimeric form of BACE1, an indicator of active BACE1 (PMID: 15247262, PMID: 15485862). We found that p25 expression level, BACE1 activity, and BACE1 dimer formation were not altered, indicating that BACE1 activity is not affected by SPIN90. We have included these results to Supplementary Figure 3, along with accompanying text.
Supplementary Figure 3
Supplementary Figure 4
- Authors did employ two different strategies (Area in Fig.1A-D, count numbers in Fig.1E-F) to measure Aβ plaques. Scientific reasoning is kindly asked and should be clearly mentioned in the text.
Response2- We apologize for this confusion based on our use of two different strategies to analyze Aβ deposition: immunohistochemistry and thioflavin S staining. The methods section now describes these analytic strategies.
Figure 1 A-D describes the results of immunohistochemistry (IHC) using DAB staining. This method was used to detect all Aβ signals in both intracellular and extracellular areas of the brain, such was the hippocampus and subiculum. In mice aged 2 months, the areas positive for Aβ were relatively small. In older mice, however, the areas of Aβ deposition were enlarged such that they partial overlapped or connected with each other. This resulted in ambiguity and inaccuracy in precisely quantifying the number of loci of Aβ deposition, reducing the ability to determine the correlation between the number of areas of Aβ deposition and age. Thus, the IHC experiments are reported as Aβ-positive areas, not as Aβ-positive loci. Thioflavin, however, stains only extracellular Aβ deposits, providing clearer fluorescence images than DAB-stained IHC images. This allowed quantification of the number of Aβ deposits.
- How is the counting methodology for the Aβ plaques? (e.g. what is the lowest threshold size limit of clusters to be considered?
Response3- To count Aβ deposits, we first had to determine the signals that represented actual Aβ deposits. We utilized two criteria to evaluate Aβ deposition. The first was the intensity of the signal, with intensities >86 arbitrary units defined as positive for Aβ deposition. The second was the shape of the image. After first filtering, we chose circular- and oval-like shapes, not long bars or other unspecified shapes, as positive for Aβ deposition. Thus, after application of both filters, the lowest threshold for Aβ deposition was 6 pixels, equivalent to less than 72 µm2.
Also, have authors happened to see any difference in size distribution of Aβ plaque clusters between 5xFAD SPIN90wt and 5xFAD SPIN90ko (Fig.1E-F)?
Response3- As suggested by this reviewer, we analyzed the sizes of Aβ deposits in 5xFAD-SPIN90 WT/KO mouse brains. Compared with the sizes of Aβ deposits in 5xFAD-SPIN90 WT mouse brains, the sizes of Aβ deposits in the hippocampus and subiculum of 5xFAD-SPIN90 KO mouse brains were decreased to ~18.5% and ~23%, respectively. These results are now included in Supplementary Figure 1 and its accompanying text.
Supplementary Figure 1
- It has been previously shown that APP has functions of the mediation of post-synaptic density composition and activity (PMID: 19164281). As authors did show in this study that SPIN90 localize at presynapses and synapses and its deficiency regulates synaptic funtions; have authors measured the APP accumulation difference in post-synaptic density between 5xFAD SPIN90wt and 5xFAD SPIN90ko?
Response4- We are aware that APP also localizes at dendrites and areas of post-synaptic density (PSD) and regulates synaptic function (PMID: 22884903, PMID: 19164281, PMID: 19726636). Moreover, studies have reported that APP is also trafficked in areas around dendrites (PMID: 26642089 PMID: 23931995).
As mentioned by this reviewer, we had initially tried to measure APP accumulation in dendritic areas, including areas of PSD, but we did not detect APP accumulation in these areas. Similarly, a study by Jorda-Siquier et al. (PMID: 35076178) reported APP accumulation in presynaptic areas along with presynaptic proteins but rarely in dendrites and areas of PSD. Additional studies are needed to determine whether APP accumulates in dendrites and areas of PSD.
- What are the criteria on selection of ROIs?
Response5- The selection of ROIs to assess APP accumulation was based on general synaptic bouton size, estimated to be 1.0~1.5 µm in diameter. ROIs were selected for APP accumulation if the diameter of APP accumulation was greater than 1.5 µm.
- Did authors introduce the same 5xFAD mutations of APP to the pH-APP vector? If so, please correctly annotate in the scheme Fig.2A.
Response6- We have corrected the schematic diagram in Figure 2A.
- There is no time-lapse live-imaging data on GFP-APP and RFP-SPIN90 move together through axons in Fig.4A, as written in the text (Lines 231-232). Authors are kindly asked to provide the supporting data.
Response7- As suggested by this reviewer, we have added the video to the supplementary materials.
- Did authors test the dynamics of the axonal motility of APP whether it's affected upon RAB11 attenuation or overexpression compared to endogenous level?
Response8-
We have examined whether Rab11 status influenced APP axonal motility in 5xFAD-SPIN90WT/KO neurons. These assays were performed by introducing active (Q70L) or inactive (S25N) forms of Rab11, along with GFP-APP, into 5xFAD-SPIN90WT/KO neurons and monitoring the axonal trafficking of APP. Stationary APP was increased in both 5xFAD-SPIN90WT and KO neurons expressing the inactive form of Rab11 (S25N), although the amount of stationary APP was much higher in 5xFAD-SPIN90 KO than in 5xFAD-SPIN90 WT neurons, suggesting that Rab11 activity along with SPIN90 influenced APP trafficking. We have added these data to Supplementary Figure 7.
Supplementary Figure 7
- Via what exact mechanism, the depletion of SPIN90 reduces the Aβ plaque level? For example, does SPIN90 sequester BACE1 into APP+ RAB11 endosomes? Did authors check BACE1-APP colocalization difference between 5xFAD SPIN90wt and 5xFAD SPIN90ko?
Response9- Based on our results, we suggest the following mechanism. SPIN90 regulates APP recycling and axonal motility via axons and the internal accumulation of APP along with interactions with inactive Rab11. This may influence amyloidogenesis and synaptic functionality, suggesting that SPIN90 may be a co-modulator of amyloidogenesis by regulating the trafficking of APP. With respect to BACE1-APP modulation, we have not yet investigated the control of these molecular interactions and colocalization. However, previous studies reported that BACE1 recycling is modulated by Rab11 (PMID: 24373285) and that some endosomes contain both APP and BACE1, which are cotrafficked (PMID: 26642089, 23931995). In addition, our study revealed that the level of expression and activity of BACE1 were not affected by SPIN90. These findings suggest that colocalization of BACE1-APP may not be particularly altered, whereas the degree of APP accumulation may be associated with amyloidogenesis.

Reviewer 2 Report
SPIN90 deficiency ameliorates amyloid β accumulation by regulating APP trafficking in AD model mice
Overall, I believe the manuscript is full of fascinating information and adheres well to the goals of the journal. Accepting the paper with a very minor correction is my recommendation.
1. Although the authors included the abbreviation at the end of the paper. Please use the complete name prior to abbreviations, such as Nck (non-catalytic region of tyrosine kinase), SPIN90 (SH3 protein interacting with Nck, 90 kDa).
2. Please remove the heading of the figure “Oh et al”
3. Please provide the catalog number and manufacturer of the antibodies used for future replication.
4. Line 505: Western blotting
5. Although the authors showed the promising molecular data in SPIN90 KO mice. However, physical data, such as Morris water maze or ORT is highly recommended.
Author Response
Reviewer #2
SPIN90 deficiency ameliorates amyloid β accumulation by regulating APP trafficking in AD model mice. Overall, I believe the manuscript is full of fascinating information and adheres well to the goals of the journal. Accepting the paper with a very minor correction is my recommendation.
We thank for the reviewer’s positive comment
1.Although the authors included the abbreviation at the end of the paper. Please use the complete name prior to abbreviations, such as Nck (non-catalytic region of tyrosine kinase), SPIN90 (SH3 protein interacting with Nck, 90 kDa).
Response1- We have used the complete name to define each abbreviation.
2.Please remove the heading of the figure “Oh et al”
Response2- This heading has been removed.
3.Please provide the catalog number and manufacturer of the antibodies used for future replication.
Response3- We have added information on each of the antibodies used, including their catalog numbers and manufacturers.
4.Line 505: Western blotting
Response4- Western blotting is not mentioned on Line 505 of the manuscript. We request clarification from the reviewer.
- Although the authors showed the promising molecular data in SPIN90 KO mice. However, physical data, such as Morris water maze or ORT is highly recommended.
Response5- Although evaluation of neural behavior is of great interest, we believe that it is outside the scope of the current study. Our previous study investigated memory extinction after fear conditioning in SPIN90 KO mice (PMID: 28979184). That study showed that decay of memory extinction is modestly affected in SPIN90 KO mice, suggesting that memory in 5xFAD-SPIN90 KO mice may be improved compared with 5xFAD-SPIN90WT mice. Future studies are planned to assess neural behavior in these mice.

Round 2
Reviewer 1 Report
Authors have addressed all my questions and comments.
Just a reminder: Authors should also mention the counting methodology of the Aβ plaques, and criteria on selection of ROIs in the methods section. Also, as authors responded that they had initially tested measuring APP accumulation in dendritic areas, including areas of PSD, but they did not detect APP accumulation in these areas; I believe that this piece of data is important for the story, so this negative finding should also be added as a supplementary figure.
Author Response
Reviewer 1
Just a reminder: Authors should also mention the counting methodology of the Aβ plaques, and criteria on selection of ROIs in the methods section.
Response: We have mentioned the counting methodology (line 420-423) and the ROI selection criteria (line 426-428) in the method section of “4.9 image analysis”
Also, as authors responded that they had initially tested measuring APP accumulation in dendritic areas, including areas of PSD, but they did not detect APP accumulation in these areas; I believe that this piece of data is important for the story, so this negative finding should also be added as a supplementary figure.
Response: We have added this result in supplementary figure 7 as the reviewer recommended.